# The GGDEF-EAL protein CdgB from *Azospirillum baldaniorum* Sp245, is a dual function enzyme with potential polar localization

Víctor I. Viruega-Góngora[1], Iris S. Acatitla-Jácome[1], David Zamorano-Sánchez[2], Sandra R. Reyes-Carmona[1], María L. Xiqui-Vázquez[1], Beatriz Eugenia Baca[1], Alberto Ramírez-Mata[1]*

1 Laboratorio de la Interacción bacteria-planta, Centro de Investigaciones en Ciencias Microbiológicas, Benemérita Universidad Autónoma de Puebla, Puebla Pue, México, 2 Programa de Biología de Sistemas y Biología Sintética, Centro de Ciencias Genómicas, Universidad Nacional Autónoma de México, Cuernavaca, México

* alberto.ramirez@correo.buap.mx

**Data Availability Statement:** All relevant data are within the paper and its Supporting Information files.

## Abstract

*Azospirillum baldaniorum* Sp245, a plant growth-promoting rhizobacterium, can form biofilms through a process controlled by the second messenger cyclic diguanylate monophosphate (c-di-GMP). *A. baldaniorum* has a variety of proteins potentially involved in controlling the turnover of c-di-GMP many of which are coupled to sensory domains that could be involved in establishing a mutualistic relationship with the host. Here, we present *in silico* analysis and experimental characterization of the function of CdgB (AZOBR_p410089), a predicted MHYT-PAS-GGDEF-EAL multidomain protein from *A. baldaniorum* Sp245. When overproduced, CdgB behaves predominantly as a c-di-GMP phosphodiesterase (PDE) in *A. baldaniorum* Sp245. It inhibits biofilm formation and extracellular polymeric substances production and promotes swimming motility. However, a CdgB variant with a degenerate PDE domain behaves as diguanylate cyclase (DGC). This strongly suggest that CdgB is capable of dual activity. Variants with alterations in the DGC domain and the MHYT domain negatively affects extracellular polymeric substances production and induction of swimming motility. Surprisingly, we observed that overproduction of CdgB results in increased c-di-GMP accumulation in the heterologous host *Escherichia coli*, suggesting under certain conditions, the WT CdgB variant can behave predominantly as a DGC. Furthermore, we also demonstrated that CdgB is anchored to the cell membrane and localizes potentially to the cell poles. This localization is dependent on the presence of the MHYT domain. In summary, our results suggest that CdgB can provide versatility to signaling modules that control motile and sessile lifestyles in response to key environmental signals in *A. baldaniorum*.

**Funding:** Consejo Nacional de Ciencia y Tecnología (CONACYT), Grant INFRA-2014-01-225923, and the financial support of Vicerrectoría de Investigación y Estudios de Posgrado (VIEP). The funders had no role in study design, data collection and analysis, decision to publish, or preparation of the manuscript.

**Competing interests:** The authors have declared that no competing interests exist.

## Introduction

The second messenger cyclic diguanylate monophosphate (c-di-GMP) plays a central role in microbial cellular behaviors in response to the environment with ecological importance such as the production of extracellular polymeric substances (EPS), biofilm formation, chemotaxis, and several forms of motility, such as swimming, and swarming [1–4]. It is now well accepted that high intracellular levels of c-di-GMP inhibit motility and promote biofilm formation [5–7].

Intracellular c-di-GMP levels are controlled by the synthesizing activity of diguanylate cyclases (DGCs), which are GGDEF domain-containing proteins, and by the degrading activity of phosphodiesterases (PDEs) harboring EAL or HD-GYP domains [8]. Many of these proteins have distinct N-terminal sensing and signaling domains, suggesting that their activities in c-di-GMP turnover respond to various intra and extracellular signals. Examples of these sensing partner domains, are the Per-Arnt-Sim (PAS) [9], GAF [10], and MHYT [5] domains, among others, which could function as sensors or signal transmitters to the catalytic component of the molecule, regulating its enzymatic activity. Furthermore, approximately half of the many thousands of these enzymes annotated in protein databases are also integral membrane proteins, often with sensor domains located within the bacterial periplasm, and a DGC or PDE domain located in the cytoplasm [11]. Bioinformatic studies revealed that several *Azospirillum* genomes contain an average of 35 genes presumptively involved in c-di-GMP metabolism. A number of these proteins were predicted to have transmembrane domains, suggesting that they might be anchored to the cytoplasmic membrane [12].

*Azospirillum baldaniorum* Sp245 previously named *A. brasilense* Sp245 [13] is a plant growth-promoting bacterium (PGPB) used worldwide to promote sustainability in agriculture [14, 15]. This alpha proteobacterium inhabits the rhizosphere and soil, possess endophytic ability, and is known to promiscuously colonize the root surfaces of a wide variety of plants [14]. These bacteria can increase the yield of important crops growing in various soils and climatic regions [15]. The bacterium must cope with a very competitive environment in the rhizosphere and soils. It is capable of forming biofilms on wheat root surfaces [16], and use c-di-GMP signaling modules to promote this social behavior [17]. The genome of *A. baldaniorum* Sp245, has 35 genes that encode 25 proteins with a single GGDEF domain, 5 with an EAL domain, and 10 that carry both domains. So far, only two have been characterized: CdgC is a functional DGC involved in biofilm formation, EPS production, and plant colonization [18], and CdgD, a DGC-PDE hybrid protein, which is involved in motility, biofilm formation, and plant colonization through its c-di-GMP synthesizing activity [19].

The goal of this study was to further uncover c-di-GMP-signaling modules that could regulate sessile and motile lifestyles in *A. baldaniorum* Sp245. Here we report our findings regarding the characterization of the product of *cdgB*, a GGDEF-EAL hybrid protein. Our results revealed that CdgB has both DGC and PDE functions under different conditions and a MHYT domain that might be involved in the polar localization of CdgB. The identification and characterization of these types of c-di-GMP metabolic enzymes with dual activity, such as CdgB, are currently understudied but they will provide insights into c-di-GMP regulated processes related to environmental adaptations needed in the interaction of Plant Growth Promoting Rhizobacteria with their host plant.

## Materials and methods

### Protein sequence analysis, structure modeling and computational docking

The primary sequence of CdgB was analyzed with the databases SMART [20], Pfam [21], and Protter [22] to predict functional domains and features relevant to its cellular localization.

Sequence alignments were performed using Clustal Omega [23] and STRAP [24]. The structural model for PAS-GGDEF-EAL of CdgB for schematic representation was built by I-TAS-SER and SwissModel servers [25, 26] taking as reference the structure of RbdA (Protein Data Bank [PDB]:5XGB). The structures of the GGDEF and EAL domains for molecular coupling analyses were built by the SwissModel server taking as reference the structures of RbdA (Protein Data Bank [PDB] code 5XGD) [27] for the GGDEF domain and MucR ([PDB] code 5M1T) [28] for the EAL domain, both from *P. aeruginosa*. The quality of the models was analyzed on the Molprobity server [29] and a subsequent structural minimization was carried out in UCSF Chimera 1.10 software [30]. The GTP and c-di-GMP substrates were taken from the PubChem database [31], to which hydrogens were added and structural minimization was performed for each in Avogadro 2.0 [32]. PDBQT files were generated in the AutodockTools tool, where $Mg^{2+}$ ion uploads were added with the SwissModel server. Molecular coupling analyses were carried out on the AutoDockTool4 platform [33]. Finally, the visualization and preparation of figures were carried out in the UCSF Chimera 1.10 program.

## Bacterial strains, plasmids, primers, media, and growth conditions

The strains, plasmids, and primers used in this work are described in **Table 1**. *Escherichia coli* strains were grown at 37˚C in Lysogeny broth (LB), and agar (1.5% w/v) was added for solid medium. Antibiotics were added, when necessary, at the following concentrations: ampicillin (Ap), 100 μg/mL; chloramphenicol (Cm), 50 μg/mL; gentamicin (Gm), 15 μg/mL; kanamycin (Km), 20 μg/mL. *A. baldaniorum* Sp245 strains were grown in K-malate minimal medium, K-lactate minimal medium, Congo red (CR) medium, LB* (2.5 mM $MgCl_2$ and 2.5 mM $CaCl_2$) medium or NFB* medium [18]. For agar plates (agar 1.5% w/v) Gm 30 μg/mL or Km 50 μg/mL were added for plasmid or transconjugants selection.

## Construction of plasmids and variant strains

Isolation of genomic and plasmid DNA used for DNA restriction enzyme digestion, electrophoretic agarose analysis, and transformation assays was carried out according to standard protocols [38].

The Δ*cdgB* mutant strain was constructed by replacing the coding region of *cdgB* with a kanamycin resistance cassette as previously described [17]. The DNA fragments flanking *cdgB* (GenBank accession number WP_014199675) were amplified by PCR using the primers Fkpnaz88 and Rxhoaz88 to generate the upstream A fragment of 992 bp, and the primers Fspeaz90 and Rsacaz90 to generate the downstream B fragment of 993 bp, each of which were subsequently cloned in the pGEM-T Easy vector, through TA cloning (Promega, Madison, WI, USA), respectively and transformed in *E. coli* DH5α to obtain the corresponding pGEM-A and pGEM-B constructs. The A fragment was excised with *Kpn*I and *Xho*I, then was ligated into pJMS-Km suicide vector previously digested with the same restriction enzymes to yield the plasmid pJMS-Km-FA which was subsequently transformed into *E. coli* DH5α. Both pGEM-B and pJMS-Km-FA were digested with *Spe*I and *Sac*I restriction enzymes and ligated to generate the pMMS construct, which contains the Δ*cdgB*::*km*^r fragment. The pMMS construct was mobilized into *A. baldaniorum* Sp245 by biparental mating using *E. coli* S17.1 as a donor strain. Transconjugants were screened on K-lactate minimal medium with kanamycin 50 μg/mL. The mutation of interest in single colonies that were resistant to Km was further confirmed by PCR and DNA sequencing analysis.

The *cdgB* overexpression constructs, were generated using the pMP2444 broad host-range plasmid [36]. The full-length ORF of *cdgB* gene, was amplified using the Forf89 and Rorf89 primers and subsequently cloned into pGEM-TEasy to obtain the construct pGEM-*cdgB*. The

**Table 1. Strains, plasmids, and primers.**

| Strain or Plasmid | Relevant Characteristics | Source or Reference |
|---|---|---|
| **Strains** | | |
| *Escherichia coli* | | |
| DH5α | *fhA2 Δ(argF-lacZ)* U169 *phoA glnV44 Φ80 Δ(lacZ)* M15 *gyrA96 recA1 thi-1 hsdR*17 | Thermo Fisher Scientific |
| S17.1 | *recA*, *thi*, *pro*, *hsdR*-M + RP4-2-Tc::Mu::Km:Tn7 | [34] |
| *Azospirillum baldaniorum* | | |
| Sp245 | Wild-type strain | [35] |
| Δ*cdgB* | *cdgB* gene deletion mutant derived from Sp245, Km$^r$ | This study |
| **Plasmids** | | |
| pGEM-TEasy | Cloning vector *f1 or*i, Ap$^R$, *lacZa*, Promoter: T7, *lac*, Sp6 | Promega |
| pGEM-A | pGEM-TEasy containg a fragment of 992 bp from 5' Flank of *cdgB* gene | This study |
| pGEM-B | pGEM-TEasy containg a fragment of 992 bp from 3' Flank of *cdgB* gene | This study |
| pGEM-*cdgB* | pGEM-TEasy with the *cdgB* gene | This study |
| pGEM-*cdgB*::*egfp* | pGEM-TEasy with a transcriptional fusion *cdgB*::*egfp* | This study |
| pGEM-*cdgB*-2 | pGEM-TEasy with the *BamH1* and *EcoR1*overhangs sites flanking *cdgB* gene | This study |
| pJMS-Km | Suicide vector derivative from pSUP202, Ap$^r$, Tc$^r$, Km$^r$ | This study |
| pJMS-Km-FA | Suicide vector derivative of pJMS-Km carrying the fragment 5' -3' of 992 bp | This study |
| pMMS | Suicide vector derivative from pJMS-Km-FA harboring *ΔdgcB*::Km$^r$ | This study |
| pMP2444 | Host-range vector carrying the eGFP (Enhancer Green Fluorescent Protein) Gm$^R$ | [36] |
| pMP-*cdgB* | Derivative from pMP2444 harboring the *cdgB* gene 45 pb before *egfp* gene | This study |
| pMP-*cdgB*$_{ΔMHYT}$ | Derivative from pMP+*cdgB* harboring the Δ(806bp) MHYT-*cdgB* | This study |
| pMP-*cdgB*$_{SGDEF-SGKEF}$ | Derivative of pMP+*cdgB* where aspartic acid is replaced by lysine on the GGDEF motif (SGDEF to SGKEF) of CdgB | This study |
| pMP+*cdgB*$_{EAL-AAL}$ | Derivative of pMP+*cdgB* where glutamic acid is replaced by alanine on the EAL motif (EAL to AAL) of CdgB | This study |
| pMP-*cdgB*::*egfp* | Derivative from pMP2444 with a translational fusion *cdgB*::egfp | This study |
| pGEX-4T-1 | Expression vector, tac promoter, GST tag, Ap$^r$ | Promega |
| pGEX-*cdgB* | Plasmid derived from pGEX-4T-1 containing the ORF of *cdgB* gene, Ap$^r$ | This study |
| pGEX-*cdgB*$_{SGDEF-SGKEF}$ | Derivative of pGEX-*cdgB* where aspartic acid is replaced by lysine on the GGDEF motif (SGDEF to SGKEF) of CdgB | This study |
| pGEX-*cdgB*$_{EAL-AAL}$ | Derivative of pGEX-*cdgB* where glutamic acid is replaced by alanine on the EAL motif (EAL to AAL) of CdgB | This study |
| pDZ-119 | c-di-GMP biosensor, Cm$^r$ | [37] |
| **Primer** | **Sequence 5´-3´** | **Reference** |
| Fkpnaz88 | GATA**GGTACC**AATGAACCGGAACGACCTCAG | This study |
| Rxhoaz88 | TAAG**CTCGAG**GCCGCTTGATCCGATTACCTT | This study |
| Fspeaz90 | GATA**ACTAGT**CACGGACGTTTTTCGCGG | This study |
| Rsacaz90 | TAA**GGAGCTC**AAGCACGACCTGTTCGTCT | This study |
| Forf89 | GATACCCGGGATGCGTGTGTATGCCTGC | This study |
| Rorf89 | TAAGAAGCTTTCACGCCGGCTCGTAGAAG | This study |
| *cdgB*-F-24 | TCGGATCC ATCGGATCAAGCGGCGTCGA | This study |
| *cdgB*-R-24 | GCCCTTGCTCACCATCGCCGGCTCGTAGAAGCGTG | This study |
| *gfp*-F-24 | TTCTACGAGCCGGCGATGGTGAGCAAGGGCGAGGA | This study |
| *gfp*-R-24 | AGTCTAGA TTACTTGTACAGCTCGTCCATGCCG | This study |
| *cdgB*-F-int | GGTACCGACCGGCTGCTGCAGGAGAT | This study |
| *gfp*-ext | TACGTATTACTTGTACAGCTCGTCCATGCCG | This study |
| B*cdgB*-F | CAGGATCCGCCATCGTCGACCAGCGGCT | This study |
| E*cdgB*-R | ACGAATTCTCACGCCGGCTCGTAGAAGC | This study |
| GGDEF-F | GGGCTGAGCGGCAAAGAGTTCGCCGTG | This study |

(*Continued*)

**Table 1.** (Continued)

| Strain or Plasmid | Relevant Characteristics | Source or Reference |
|---|---|---|
| GGDEF-R | GACGGCGAACTCTTTGCCGCTCAGCCG | This study |
| EAL-F | ATCCTGGGCTTCGCGGCGCTGGTGCGC | This study |
| EAL-R | GCGCACCAGCGCCGCGAAGCCCAGGAT | This study |

Ampicillin = Ap$^r$; Gentamycin = Gm$^r$; Chloramphenicol = Cm$^r$; Kanamycin = Km$^r$; Tetracycline = Tc$^r$.

pGEM-*cdgB* and pMP2444 plasmids were digested with the *Eco*R1 restriction enzyme. The *EcoRI* digested fragments corresponding to *cdgB* and linearized pMP2444 were ligated to generate pMP-*cdgB*. The desired 5'–3' orientation of the insert (*cdgB* expressed under the *lac* promoter) was confirmed by restriction analysis with *Hin*dIII. Competent *E. coli* S17.1 cells were transformed with the pMP-*cdgB* plasmid. Transformed cells were used as donors in biparental matings to transfer pMP-*cdgB* to *A. baldaniorum* Sp245. The point mutation on the GGDEF motif (D456K) was introduced by inverted PCR using the Q5 Site-Directed Mutagenesis kit (New England BioLabs) following the manufacturer's instructions (using the primer pair GGDEF-F and GGDEF-R). The resulting pMP-*cdgB*$_{SGDEF-SGKEF}$ plasmid was introduced into *E. coli* S17.1 and subsequently transferred to *A. baldaniorum* Sp245 by biparental conjugation. The same strategy was used for introducing the point mutation in the EAL motif (E580A) (using the primer pair EAL-F and EAL-R). The resulting pMP-*cdgB*$_{EAL-AAL}$ plasmid was introduced into *E. coli* S17.1 and subsequently transferred to *A. baldaniorum* Sp245 through conjugation.

All plasmid constructs were sequenced to confirm the correct sequence of *cdgB* and its desired orientation (downstream and under the control of the *lac* promoter). These constructs do not have the gene that produces the LacI repressor, hence the Plac promoter is constitutively active.

To generate a *cdgB*::e*gfp* translational fusion we used a PCR fusion approach designed by Yang et al. [39]. Briefly, the *cdgB* gene, without a stop codon, was amplified with primers *cdgB*-F-24 and *cdgB*-R-24, and the *egfp* reporter gene was amplified using *gfp*-F-24 and *gfp*-R-24. The primer *cdgB*-R-24 and *gfp*-F-24 have compatible overhangs that allow fusing the amplicons through PCR amplification using the primer pair *cdgB*-F-24- *gfp*-R-24. The fused PCR product and plasmid pMP2444 were digesting with *Bam*H1 and *Xba*I restriction enzymes, and ligated to generate the plasmid pMP-*cdgB*::*egfp*. *E. coli* S17.1 competent cells were transformed with pMP-*cdgB*::*egfp* and a transformant was used to transfer the plasmid to *A. baldaniorum* Sp245 by conjugation to as previously described [16].

The MHYT domain was deleted by digesting the pMP-*cdgB*::*egfp* construct with *Sal*I. The digested fragment of 7113 bp that lack the MHYT domain was ligated to produce the pMP-*cdgB*$_{\Delta MHYT}$ plasmid, which was introduced to *E. coli* S17.1 by chemical transformation. The plasmid was transferred by conjugation to *A. baldaniorum* Sp245. The *cdgB*$_{\Delta MHYT}$ deletion allele was sequenced to verify that the open reading frame was not shifted.

To generate the pGEX-*cdgB* construct, *cdgB* was amplified by PCR using primers B*cdgB*-F and E*cdgB*-R and inserted in the pGEM-TEasy plasmid, obtaining the pGEM-*cdgB*-2 construct. The pGEM-*cdgB*-2 and pGEX-4T1 plasmids were digested with the *Bam*H1 and *Eco*R1 enzymes, the digested fragment containing *cdgB* was ligated to the linearized pGEX-4T1 plasmid to generate pGEX-*cdgB*. To obtain the variants with point mutations on GGDEF and EAL motifs (pGEX-*cdgB*$_{SGDEF-SGKEF}$ and pGEX-*cdgB*$_{EAL-AAL}$) we used the same strategy above described. We used the same primer set (GGDEF-F, GGDEF-R and EAL-F, EAL-R). The sequences of the three inserts were analyzed through Sanger sequencing. All constructs

(pGEX-*cdgB*, pGEX-*cdgB*$_{SGDEF-SGKEF}$ and pGEX-*cdgB*$_{EAL-AAL}$) were used to transform competent *E. coli* S17.1 cells harboring the c-di-GMP biosensor pDZ-119 [37].

## Analysis of growth curves

To determine possible effects of plasmid pMP-*cdgB* or Δ*cdgB* deletion mutant on growth rates, growth curves were performed and compared with strains carrying the empty vector pMP2444 or WT strain. Overnight cultures were diluted to an optical density at 600 nm (OD$_{600}$) of 0.01 in 100 ml Erlenmeyer flasks (3 replicates per strain) containing 25 ml of NFB* medium supplemented with Gm, when it was necessary for plasmid selection. Cultures were incubated in a rotary shaker (150 r.p.m.) at 30˚C and OD$_{600}$ measured every 2 hours, using an EON microplate spectrophotometer (BioTek, Winooski, VT, USA) at 595 nm.

## Biofilm formation assay

The biofilm formation assay was performed as previously described [17, 40, 41]. Briefly, *A. baldaniorum* Sp245 and the *A. baldaniorum* derived strains (*A. baldaniorum* Δ*cdgB*, *A. baldaniorum* Δ*cdgB*+pMP-*cdgB*, *A. baldaniorum*+*cdgB*, *A. baldaniorum*+*cdgB*$_{ΔMHYT}$, *A. baldaniorum*+*cdgB*$_{SGDEF-SGKEF}$, *A. baldaniorum*+*cdgB*$_{EAL-AAL}$, *A. baldaniorum*+*cdgB*::*egfp*) were grown overnight at 30˚C in LB* medium. Next, 7.5 mL of NFB* medium supplemented with KNO$_3$, in borosilicate round-bottom tubes were inoculated with 75 μL of overnight cultures grown in LB* diluted to an OD$_{600}$ of 2.0. After inoculation, the cultures were incubated for 5 days at 30˚C under static conditions. Biofilms were stained with a 0.5% (w/vol) crystal violet solution for 30 min, and then rinsed with distilled water. Bound crystal violet was solubilized with 2 mL of acetic acid 33% (v/v), and crystal violet concentration was determined using 96-wells microassay plates read in an EON microplate spectrophotometer (BioTek, Winooski, VT, USA) at 595 nm. Crystal violet measurements were normalized by total protein concentration measured with the Bradford method. The results are from three independent experiments with three biological determinations.

## Determination of extracellular polymeric substance production

To analyze EPS production, *A. baldaniorum* and derivative strains were grown in LB* and diluted to obtain bacterial suspensions with an OD$_{600}$ of 1.2–1.4 in NFB* medium supplemented with KNO$_3$ and grown for five days at 30˚C, under static conditions. Next, cultures were centrifuged at 10,000 g, the pellets were suspended in 1 ml NFB* media, as described above, and a 0.005% (w/v) Congo Red (CR) colorant solution (Sigma-Aldrich, Chemical) was added to achieve a 40 μg/ml concentration. The cells were incubated with agitation (200 rpm) for two hours. Afterward, CR bound to cells was quantified as previously described [17, 19]. CR measurements were normalized by total protein concentration measured with the Bradford method. The results are of three independent assays with three biological determinations.

## Motility assay

The swim motility assay was performed as previously described [19, 42]. Briefly, bacteria were grown in LB* medium at 30˚C until reaching 5x10$^6$-5x10$^7$ CFU/mL, afterwards 5 μL of the culture were spotted over semisolid minimal K medium supplemented with malate, succinate, or proline (10 mM) as a carbon source and containing 0.25% (w/v) agar. The size of the bacterial motility ring was measured in cm after incubation at 30˚C for 48 h.

## Relative quantification of c-di-GMP accumulation using a genetic biosensor

C-di-GMP levels were analyzed using the riboswitch-based dual fluorescence reporter system as described previously [18, 19, 43]. This biosensor expresses AmCyan and TurboRFP, both fluorescent proteins (green and red, respectively), from the same constitutive promoter. TurboRFP production is inhibited at low levels of c-di-GMP due to the presence of c-di-GMP-binding riboswitch. For this purpose, the constructs pGEX-CdgB, pGEX-*cdgB*$_{SGDEF-SGKEF}$, pGEX-*cdgB*$_{EAL-AAL}$, pGEX-CdgA (positive control) [17] and the empty vector pGEX-4T-1 (negative control) were used to transformed a competent *E. coli* S17.1 strain containing the pDZ-119 vector [37]. All bacterial strains were grown in LB medium at 30˚ to an OD$_{600}$ of approximately 0.6. Afterwards, 0.1 mM IPTG was added to the bacterial culture, and these were further incubated for 24 h for induction of protein expression. Cultures were concentrated 10-fold and resuspended in water. c-di-GMP production was analyzed macroscopically relating color intensity with the production of the second messenger. Microscopic assessment of c-di-GMP was performed using a Nikon Eclipse TE2000U fluorescence microscope. A drop of the induced culture was deposited on a coverslip and covered with a 1% agarose pad. The excitation and emission of the calibrator AmCyan fluorophore were recorded at 457 and 520 nm, respectively. The reporter TurboRFP fluorophore was excited at 553 nm, while its emission was measured at 574 nm. Images obtained were edited with the Nikon NIS Elements. Merge images represented the overlay of the fluorescence images AmCyan green, TurboRFP red and both yellow. The relative fluorescence intensity (RFI) was calculated as the ratio between the TurboRFP and AmCyan fluorescence intensities and is directly proportional to c-di-GMP levels, as analyzed using ImageJ software. The RFI values represent the standard deviations of three biological replicates, and significant differences are indicated at $^*$ $P < 0.05$ according to Student´s t-test by SigmaPlot as previously described [19, 44].

## Microscopy studies

To visualize bacteria by fluorescence and confocal laser scanner microscopy (CLSM), *A. baldaniorum* Sp245 and its derivative strains were grown in 5 mL of NFB$^*$ medium at 30˚C, with agitation (150 rpm) for 18 h. Then, 1 mL of each culture was centrifuged at 8,000 rpm for 2 min, and each pellet was suspended in 100 μL of FM4-64FX dye (10 μg/mL) (Thermo Fisher Scientific) in phosphate buffer saline (PBS) and maintained for 10 min at 4˚C to stain the lipids of the membrane. FM4-64FX fluorescence was observed with a microscope (TE 2000U; Nikon, Tokyo, Japan) with a 100x objective lens (Plan Oil immersion) and photographed with a DS-QilMc camera (Nikon). Subsequently, 10μL of each suspension was mounted onto a glass coverslip and sealed with a 1% (w/v) agarose plug. When necessary, we also used DAPI to stain nucleoids. For video recordings, the samples were viewed with an Eclipse Ti-E C2+ confocal laser scanning microscope (Nikon) with a 60X objective lens (Plan Apo VC, water immersion). eGFP was excited at 488 nm, and its fluorescent emission was captured at 510 nm, FM4-64FX was excited at 565 nm and its fluorescent emission was captured at 734 nm, while DAPI was excited at 358 nm and its fluorescent emission was captured at 461 nm. Image slices were visualized and processed using the NIS Elements software (Nikon). The images were edited with ImageJ software (NIH, Bethesda, MD, USA) as previously described [16]. To visualize biofilms formed by the strains under study, the bacteria were grown in NFB$^*$+KNO$_3$ medium supplemented with 85 μM of Calcofluor-White colorant (CWC) (Sigma-Aldrich, United States) and inoculated onto FluoroDish glass-bottom dishes (Fisher Scientific), at 30˚C in static conditions as previously described [41].

Biofilms forming on the surfaces of dishes were recorded after 5 days and observed by CLMS. CWC was excited at 440 nm with a UV laser and its emission was captured at 500–520 nm. The samples were scanned at an x/y scanning resolution of 1,024 × 1,024 pixels. The step size in the z-direction was 0.1 μm. The image stacks were visualized and processed using NIS Elements and edited using ImageJ as previously described [41]. This analysis allowed generating a three-dimensional view of the biofilm through the measurement of signal intensity. The biofilm structure can be observed as an intensity surface plot, where the intensity of the signal represents the density of EPS produced.

## Western blotting analysis

For analysis of the expression of GFP-fusion proteins in *A. baldaniorum* Sp245, derivative strains were cultured in NFB*+KNO$_3$ medium at 30˚C to an OD$_{600}$ of 1.0. Bacteria cells were collected by centrifugation at 10,000 rpm for 5 min. The cells were resuspended in PBS, supplemented with Roche Complete Mini EDTA-free protease inhibitor, and ultrasonicated for 2 min. Cellular debris was sedimented by centrifugation (30 min. at 10,000 X g). The supernatant (soluble fraction) was obtained, and the pellet (insoluble fraction) was resuspended in PBS supplemented with detergent mix (0.5% Triton X 100; 0.1% SDS, and 0.5% 7-Deoxycholic acid sodium salt), and centrifuged at 25,000 X g for 2 h, to obtain protein membrane total extract. The total protein concentration of the respective fractions was determined using the Bradford method. The samples were resuspended in 5X loading buffer, boiled, separated on 10% SDS-PAGE, and subsequently transferred onto polyvinylidene fluoride membranes (Merck Millipore, Darmstadt, Germany) for immunoblotting using anti-GFP antibody (GFP D5.1 Rabbit mAb, HRP Conjugate, Cell Signaling Technology, Danvers, Massachusetts, USA.) HRP-DAB substrate Kit (Thermo Scientific Pierce) was used to check the target proteins on the membrane according to the manufacturer's guidelines.

## Statistical analysis

Means were compared by Student's t-test to determine statistically significant differences. Differences were considered significant at *P*-values less than 0.05.

## Results

### The domain architecture of CdgB contains multiple sensory domains and conserved GGDEF and EAL domains

CdgB (AZOBR_p410089) from *A. baldaniorum* Sp245 [45] is 794 amino acid residues long and is predicted to have seven transmembrane regions, a PAS domain, a GGDEF and an EAL domain (**Fig 1A and 1B**). The transmembrane regions conform a conserved MHYT domain which proposed to be involved in gas sensing [46, 47] (**Fig 1C**). Three MHYT motifs, characterized by conserved methionine, histidine, tyrosine, and threonine residues, are located in positions 64–67, 125–128, and 187–190 (**Figs 1B and S1**). Each MHYT motif spans two transmembrane helices, and is projected to the outer face of the cytoplasmic membrane [46]. The PAS domain faces the cytoplasm and is comprised by 107 amino acid residues from position 256 to 362. The C-terminus of CdgB resides in the cytosol and includes both the GGDEF (363–534 aa) and EAL (544–786 aa) domains (**Fig 1B**).

The primary sequence of the GGDEF domain of CdgB was aligned to the GGDEF domain of two characterized DGCs, PleD and RbdA [27, 48]. CdgB has an SGDEF motif in place of the canonical GGD[E]EF motif. This motif has been found in active DGCs in both bacteria and eukaryotes [49–51]. PleD and RbdA have an allosteric autoinhibitory site with a RXXD motif,

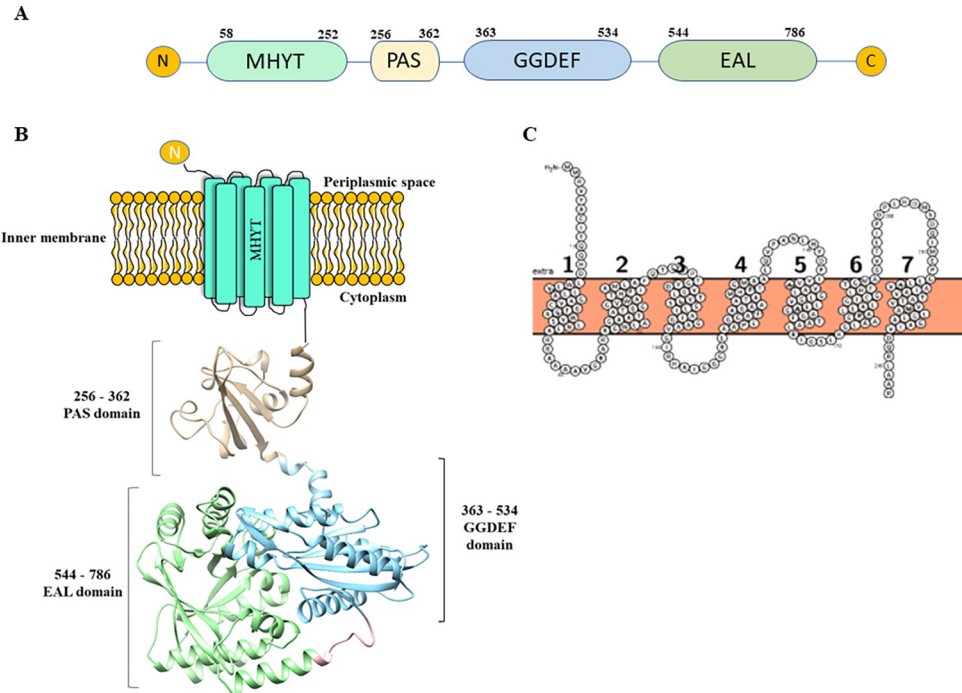

**Fig 1. Structural architecture and homology modeling of CdgB protein of *A. baldaniorum* Sp245.** (A) Schematic representation of the conserved domains of CdgB determined by the SMART database (http://smart.embl-heidelberg. de). (B) Transmembranal regions of CdgB analyzed with the Protter server. (C) Schematic representation of the 3D structure of the CdgB, PAS-GGDEF-EAL domains. The MHYT transmembrane domain was not integrated into the homology model, however, it is represented schematically into the cytoplasmic membrane cartoon. The PAS domain is in beige, the GGDEF domain is in blue, and the EAL domain is shown in green.

CdgB has a PXXD motif instead, hence it is most likely not autoinhibited by its catalytic product (**Fig 2A and 2B**).

The amino acid sequence of the EAL domain of CdgB was aligned to the sequence of the EAL domain of two well characterized PDEs, RocR and MucR [28, 52]. The alignment revealed that the EAL domain of CdgB conserve the residues involved in binding to c-di-GMP ($Y^{565}$, $Q^{566}$, $P^{567}$, $R^{584}$, $D^{698}$, $D^{720}$), $Mg^{2+}$($E^{580}$, $N^{365}$, $E^{667}$, $D^{697}$, $K^{718}$, $E^{757}$, $Q^{774}$), and for the formation and stability of loop 6 ($E^{670}$). This loop is an essential structure for dimerization [53] (**Fig 3A and 3B**).

To extend our *in silico* analysis of CdgB, we conducted tertiary structure predictions using available crystal structures of validated GGDEF and EAL domains from the Protein Data Bank repository. The GGDEF domain from CdgB was modeled using as template the crystallographic coordinates of the GGDEF domain from RbdA (PDB input: 5XGD; identity 31.84%) from *P. aeruginosa* [27]. The EAL domain from CdgB was modeled using as template the coordinates of the three-dimensional structure of the EAL domain of MucR (PDB input: 5M1T; identity 45.82%) from *P. aeruginosa* [28] (**Figs 2 and 3**).

The three-dimensional model of the GGDEF domain of CdgB was used to analyze the potential interaction with its substrates using molecular coupling analyses (molecular docking) (**Fig 2C**). Our analysis predicted an interaction of GTP with the GGDEF domain of CdgB with ΔG = -8.2 Kcal/mol (**Fig 2D**). The interaction interphase includes residues $N^{420}$, $D^{429}$, $D^{455}$. These amino acids, conserved in GGDEF domains, have been shown to be essential for the activity of DGCs [27]. The first 2 residues are important for the binding of GTP, while $D^{455}$ (located at the active site) coordinates the $Mg^{+2}$ ion to perform the nucleophilic attack on the

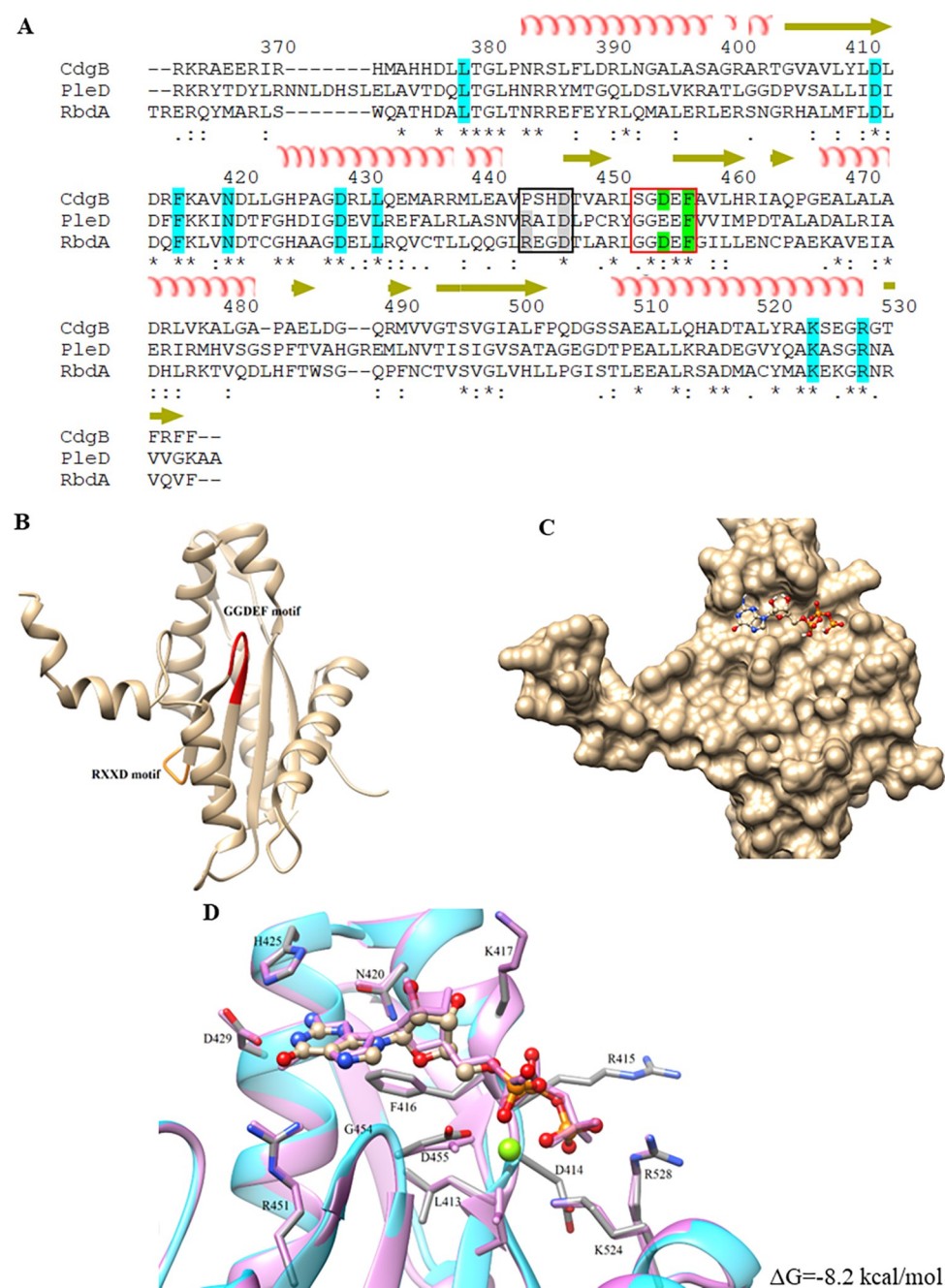

**Fig 2. CdgB GGDEF domain structure.** (A) Sequence alignment between CdgB GGDEF domain from *A. baldaniorum* Sp245 (GenBank access number CCD03171.1) and other GGDEF domains of known structure. PleD, *C. crescentus* (GenBank access number YP_002517919.1), and RbdA, *P. aeruginosa* (GenBank access number AAG04250.1). Secondary structure elements are displayed above 719 the sequences. The residues highlighted in blue are involved in the interaction with the substrate; in green, those that interact with the cofactor; the active site of the GGDEF motif is highlighted in red square, and the RXXD motif is in black square. (B) Structural model of the CdgB GGDEF monomer generated by homology modeling. The GGDEF motif is highlighted in red and the RXXD pattern is highlighted in orange. (C) Structural model of CdgB GGDEF monomer in complex with GTP. The GGDEF domain is shown in surface mode. (D) Model of the GGDEF/GTP interaction generated by molecular coupling with a ΔG = - 8.2 Kcal/mol and overlapped with the 5XGD crystal structure, corresponding to GGDEF domain from *P. aeruginosa* showed in light purple color with an RMSD of 0.539 of ligands. The Mg$^{2+}$ ion is shown in green.

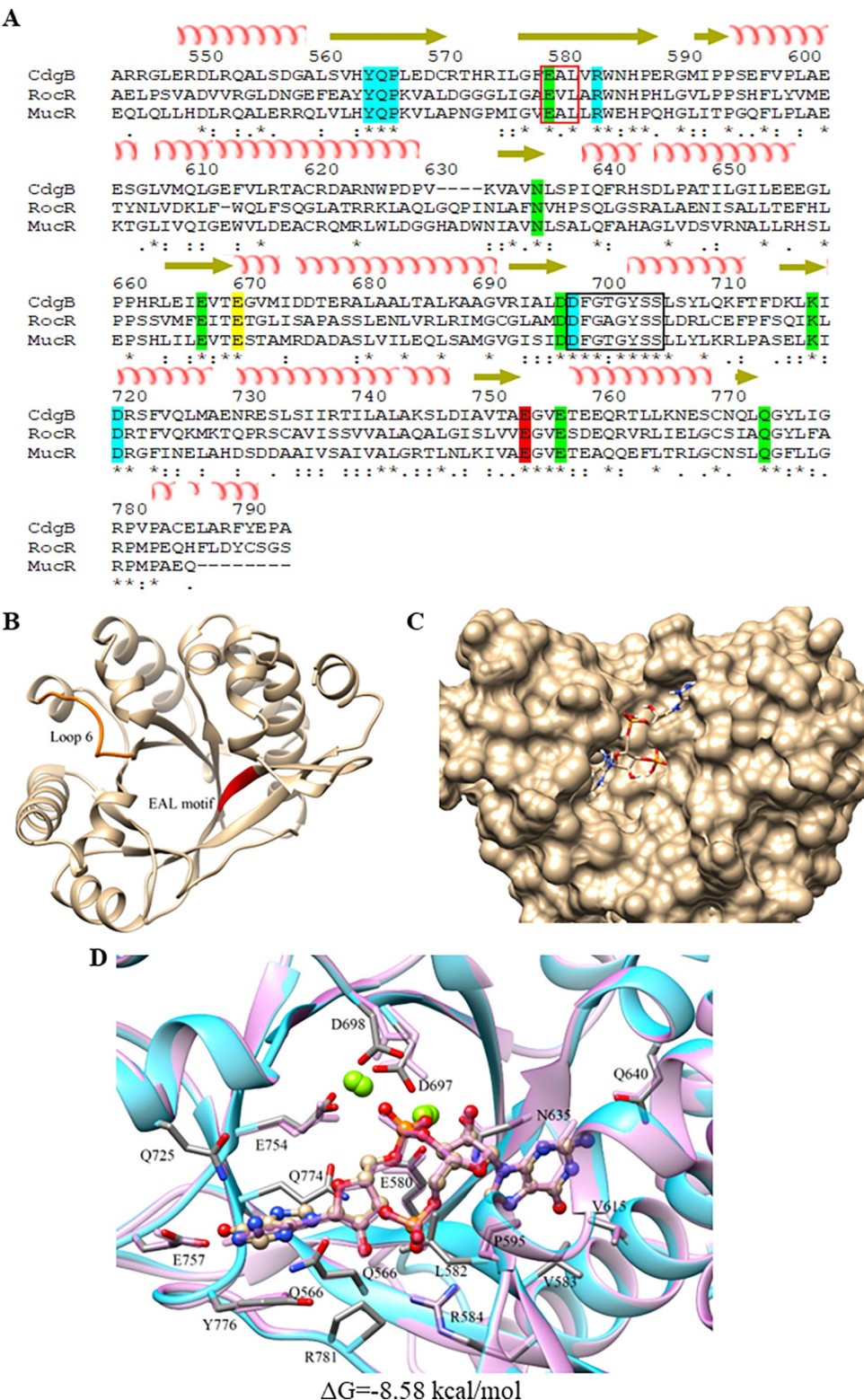

**Fig 3. CdgB EAL domain structure.** (A) Sequence alignment between CdgB EAL domain from *A. baldaniorum* Sp245 (GenBank access number CCD03171.1) and other EAL domains of known structure. RocR, (GenBank access number NP_252636.1), and MucR (GenBank access number AAG05116.1) from *P. aeruginosa*. Secondary structure elements are displayed above the sequences. The residues highlighted in blue are involved in the interaction with the substrate. Those that interact with cofactor are marked in green. The residue that stabilizes structurally the loop 6 is shown in

yellow. The catalytic residue is in red. The EAL motif is marked with a red box, and loop 6 is inside a box with black outline. (B) Structural model of the CdgB EAL monomer generated by homology modeling. The EAL motif is highlighted in red and loop 6 is highlighted in orange. (C) Structural model of the CdgB EAL monomer in complex with c-di-GMP. The EAL domain is presented in surface mode. (D) Model of the EAL domain in complex with c-di-GMP generated by molecular coupling with a ΔG = -8.58 Kcal/mol and overlapped with the 5M1T crystal structure, corresponding 744 to EAL domain from *P. aeruginosa* showed in light purple color with an RMSD of 0.757 of ligands. The $Mg^{2+}$ ions are shown in green.

alpha phosphate of GTP eliminating a pyrophosphate, resulting in the formation of c-di-GMP from another GTP molecule [48, 54]. Molecular docking analysis also predicted that the EAL domain of CdgB can bind c-di-GMP with a ΔG = -8.58 Kcal/mol (**Fig 3C and 3D**). The $N^{635}$ residue is responsible for coordinating $Mg^{2+}$ ions and is essential for the EAL domain, as its mutation was reported to significantly affect PDE activity [28, 53]. The amino acid residues $R^{584}$ and $Q^{566}$ could be involved in substrate binding by interacting with the anionic phosphate oxygen and guanine moieties of c-di-GMP [52, 54, 55]. These analyses suggest that CdgB could potentially act as a dual DGC/PDE protein.

We found *cdgB* orthologues in different species within the *Azospirillum* genus, as well as in other alpha-proteobacteria (**S2 Fig**). This could suggest that this domain architecture and potential bifunctional activity may give these bacteria a selective advantage or the ability to adapt to changes in environmental signals.

## Analyses of phenotypical consequences of alterations in *cdgB* in *A. baldaniorum* Sp245

The role of GGDEF-EAL proteins with dual activity has been poorly explored, hence we decided to investigate whether *cdgB* influences the biofilm formation, EPS production, and swimming motility of *A. baldaniorum* Sp245. To do so we generated an *A. baldaniorum* Sp245 strain with a deletion *cdgB* (*A. baldaniorum* Δ*cdgB*) and use different overexpression constructs to analyze the effect of overproducing the WT allele (pMP-*cdgB*) and mutated alleles with different point mutations that result on altered GGDEF or EAL motifs (pMP-*cdgB*SGDEF-SGKEF and pMP-*cdgB* EAL-AAL respectively), or a deletion of the MHYT domain (pMP-*cdgB*ΔMHYT), we also included the *A. baldaniorum* pMP-*cdgB*::*egfp* to analyze the stability of the CdgB::eGFP fusion. As negative controls, we analyzed strains containing the broad host-range plasmid pMP2444 (**Table 1**).

We first evaluated the growth rate of the strains of interest in NFB* media supplemented with $KNO_3$ as a nitrogen source [18, 19]. Our results showed that all the strains exhibited similar growth characteristics (**S3 Fig**). Next, we analyzed the results from biofilm formation assays. The Δ*cdgB* mutant strain, made 25% less biofilm compared to the parental strain (**Fig 4**). This could suggest that in the WT background CdgB is required for biofilm formation. Since c-di-GMP is required for biofilm formation this result would suggest that a baseline levels of expression CdgB may act as a DGC. The presence of the complementation construct pMP-*cdgB* in the Δ*cdgB* mutant strain partially restored biofilm formation compared to the WT strain with the overexpression plasmid pMP2444 (**Fig 4**). Interestingly, the overproduction of CdgB, CdgB-ΔMHYT and CdgB-SGDEF-SGKEF in the WT background of *A. baldaniorum* Sp245, resulted in decreased biofilm formation compared to both the parental WT strain and the Δ*cdgB* mutant strain. This suggests that, in the presence of an intact endogenous copy of *cdgB*, overproduction of CdgB from a plasmid has a negative effect on biofilm formation. This would indicate that at levels beyond a yet not defined threshold, CdgB may act as a PDE. To further evaluate the potential PDE activity of CdgB, we overproduced a CdgB variant with point mutations in the EAL domain that alter amino acids required for its catalytic

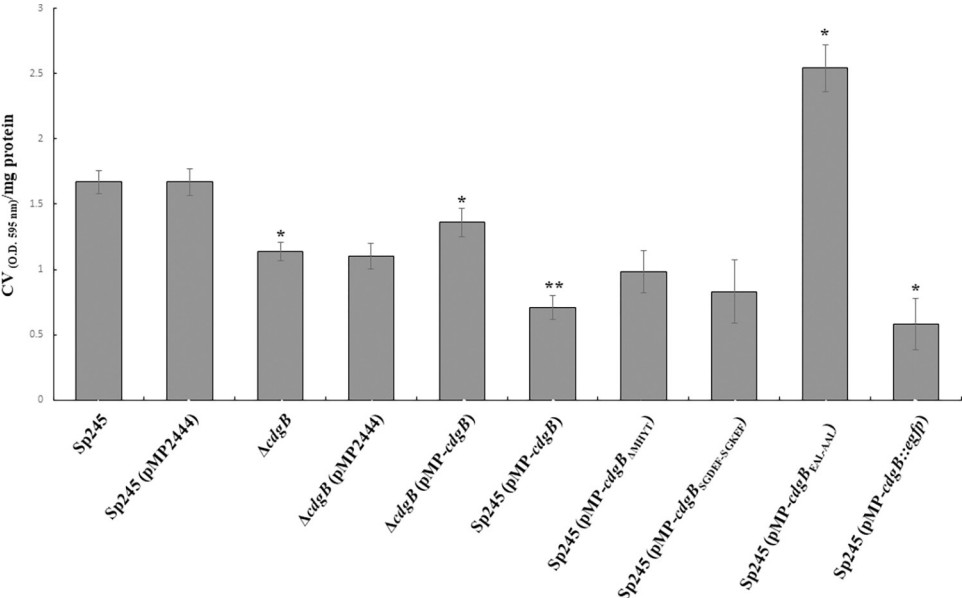

**Fig 4. CdgB alters biofilm formation in A. baldaniorum Sp245.** Graph representing the mean and standard deviation of biofilm quantification from 5 days of growth in statical conditions at 30˚C. Data represent results of three independent experiments with three biological replicates. Significant differences relative to WT (P < 0.05) are indicated by "*"and to +*cdgB* (P < 0.005) by "**" according to Student´s t-test.

activity (CdgB-EAL-AAL). The overproduction of CdgB-EAL-AAL in the WT strain, instead of negatively impacting biofilm formation as observed when overproducing WT CdgB, it promoted biofilm formation at levels higher than the control strain (**Fig 4**). This result suggests

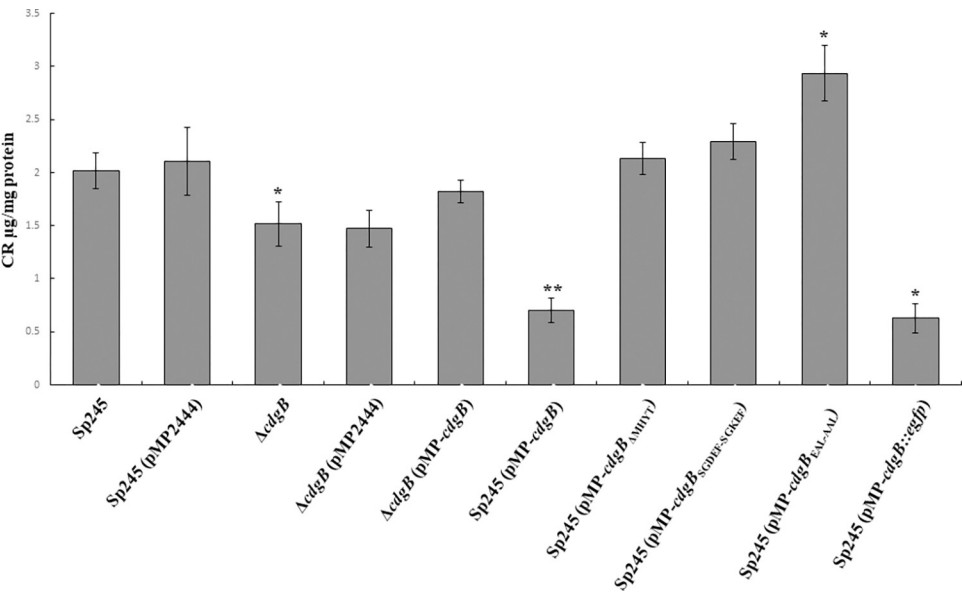

**Fig 5. CdgB alters EPS accumulation in *A. baldaniorum* Sp245.** Graph representing the mean and standard deviation of EPS quantification after 5 days of growth in statical conditions at 30˚C. Data represent results of three independent experiments with three biological replicates. Significant differences relative to WT (P < 0.05) are indicated by "*"and to +*cdgB* (P < 0.005) by "**" according to Student´s t-test.

that, when overproduced, CdgB can act as a DGC when the amino acids required for PDE activity are altered. Together these results led us to propose that CdgB is a dual-function enzyme capable of synthesizing and degrading c-di-GMP.

To further support our conclusions, we next estimated the ability of these strains to produce extracellular polymeric substance (EPS) that are likely required for biofilm formation. The strains were grown in NFB*+ $KNO_3$ and tested for their capacity to bind the colorant Congo Red (**Fig 5**), a dye that binds extracellular DNA, several exopolysaccharides, amyloid proteins and has been used as an indirect measurement of c-di-GMP production [56, 57]. In general, we observed that the biofilm phenotype of the different strains positively correlates with EPS production (**Figs 4 and 5**), except for the strains that overproduce CdgB-ΔMHYT and CdgB-SGDEF-SGKEF. We did not observe significant differences in EPS production in strains overproducing these CdgB variants compared to the control strain (no CdgB overproduction). These variants may have reduced PDE activity although more evidence is required to make that conclusion.

We further analyzed exopolysaccharide production on mature biofilms formed under static conditions in NFB* media supplemented with $KNO_3$ by staining with the dye calcofluor white (CWC) (**Fig 6**). This dye shows affinity to polysaccharides with β-1,3 and β-1,4 linkages, such as those present in capsular polysaccharides and exopolysaccharides produced by *Azospirillum* [58]. The Δ*cdgB* mutant strain and the CdgB-overexpressing-strain showed a low CWC binding, compared to WT and control strains in mature biofilms. In contrast, the strain overproducing CdgB_{EAL-AAL} produced more exopolysaccharides compared to the controls. The strain

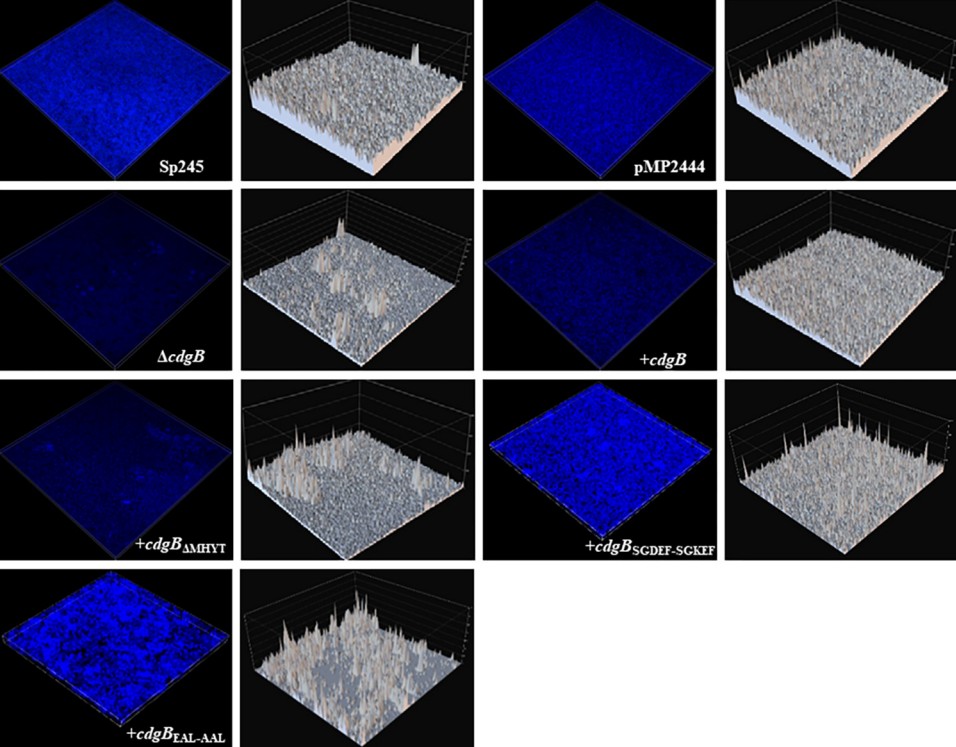

**Fig 6. Biofilm formation analysis with confocal laser scanner microscopy.** Images represent Three-dimensional reconstruction of 5-day old static biofilms; the blue fluorescence signal represents exopolysaccharides staining. Gray images are Intensity surface plots corresponding to the emitted fluorescence by CWC. The biofilm formed was directly observed using Nikon confocal microcopy C2+ with a 20X objective (CFI Plan ApVC 20X numerical aperture of 1.2). Exciting laser intensity, background level contrast, and electronic zoom were maintained at the same level.

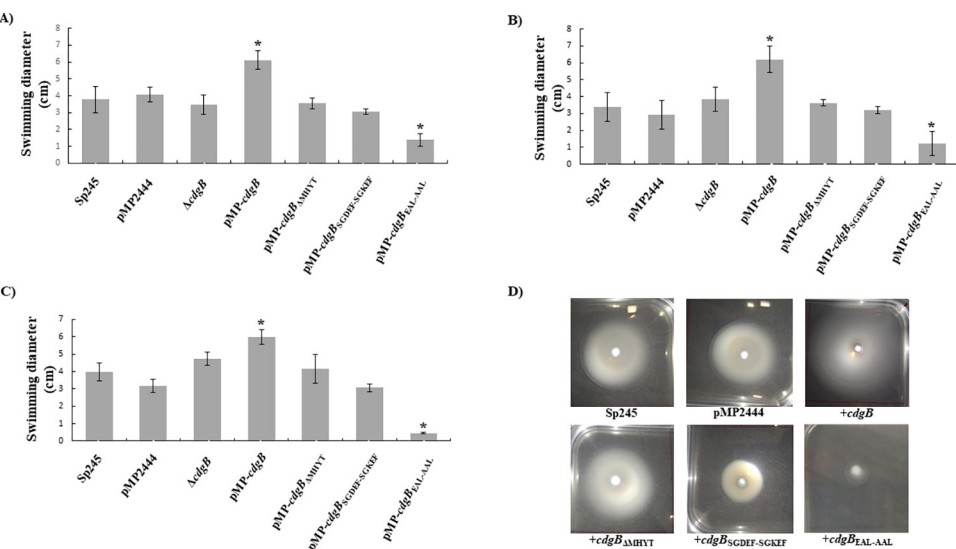

**Fig 7. CdgB alters swimming motility in *A. baldaniorum* Sp245.** Graph representing the mean of motility halos, in centimeters (cm), of the indicated strains on K-media plates with agar 0.25%, supplemented with (A) malate, (B) succinate, or (C) proline (10mM) at 30°C for 48 h. Error bars indicate standard deviations of 6 replicates per strain. (D) Representative images of motility halos from three independent experiments with two biological determinations. Significant differences relative to WT ($P < 0.005$) according to Student´s t-test are indicated by "*".

overproducing CdgB$_{SGDEF-SGKEF}$ behaved as the WT strain and the negative control, while the strain overproducing CdgB$_{\Delta MHYT}$ behaved comparable to the strain overproducing CdgB (**Fig 6**). These results further support our observations on the role of CdgB in controlling biofilm formation and EPS production, as well as for the proposed importance of EPS production on biofilm maturation in *A. baldaniorum* Sp245 [59].

Since motility is regulated by c-di-GMP in a opposite manner compared to biofilm formation and EPS production, we next evaluated the effect of overexpressing CdgB and the mutated variants on swimming motility in the presence of a variety of chemosensory signals (malate, succinate and proline). The overproduction of CdgB promoted swimming motility in the presence of any of the tested chemoattractants, while overproduction of the CdgB variant with the AAL motif had the opposite effect. Strains overproducing the CdgB variants with either the SGKEF motif or the deletion of the MHYT domain, as well as the Δ*cdgB* mutant strain, showed none or modest effects on motility (**Fig 7**). These results further suggest that CdgB can affect phenotypes associated with the levels of c-di-GMP, and hence that it is most likely a dual-function DGC-PDE. The switch in catalytic activity could depend on its abundance in *A. baldaniorum* cells or the presence of signals that inhibit its PDE activity.

## CdgB has DGC activity in *Escherichia coli*

To analyze the effect of CdgB on c-di-GMP accumulation we examined the cellular levels of c-di-GMP with the use of a c-di-GMP biosensor in the heterologous host *E. coli* S17.1 [18]. When the concentration of the second messenger increases, TurboRFP increases proportionally [37, 43]. To this end we analyzed CdgB native and CdgB variants with point mutations in the DGC domain (CdgB$_{SGDEF-SGKEF}$) and the EAL domain (CdgB$_{EAL-AAL}$), respectively. The expression of CdgB (pGEX-CdgB/ pDZ-119 strain) resulted in accumulation of red fluorescent individual cells and the formation of red fluorescent colonies (**Fig 8**). These traits were also observed when the previously characterized DGC CdgA was overproduced, but not when the empty plasmid pGEX-4T1 was used instead (**Fig 8**) [17]. The bacterial cells containing

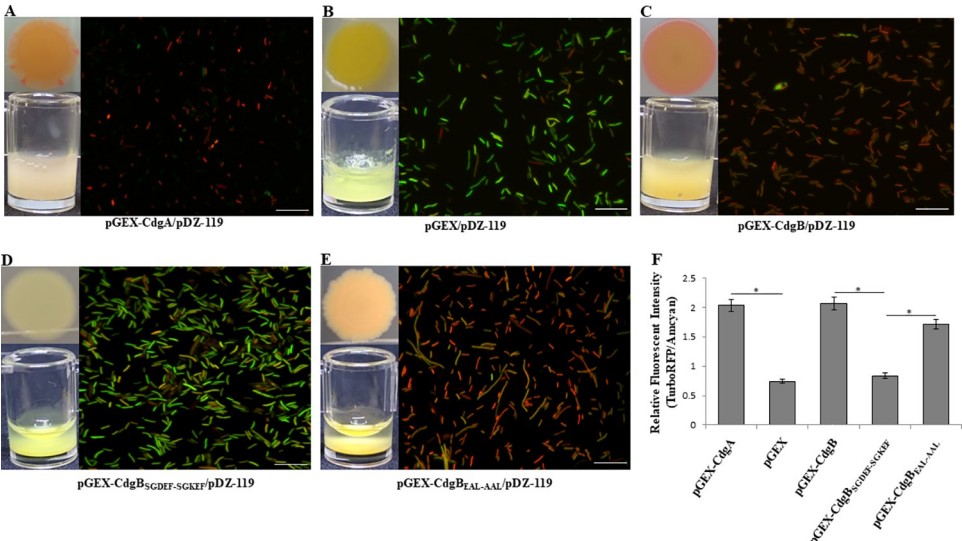

**Fig 8. CdgB promotes c-di-GMP accumulation in a heterologous host.** Representative images of macroscopic and microscopic observations of c-di-GMP levels judged by green and red coloring of colony spots and liquid concentrates as well as green and red fluorescence of individual cells from strains harboring the genetic biosensor in pDZ-119 and (A) pGEX-CdgA (DGC positive control), (B) pGEX-4T1 (negative control), (C) pGEX-CdgB, (D) pGEX-CdgB$_{SGDEF-SGKEF}$, or (E) pGEX-CdgB$_{EAL-AAL}$. (F) The relative fluorescence intensity represents the ratio between the TurboRFP and AmCyan fluorescence intensities and is directly proportional to c-di-GMP levels, as analyzed using ImageJ software. The RFI values represent the standard deviations of three biological replicates, and significant differences are indicated at $^*$($P < 0.05$) according to Student´s t-test by SigmaPlot. The white bar corresponds to 10 μm.

pGEX-CdgB$_{SGDEF-SGKEF}$ expression vector showed the same color as that of the negative pGEX-4T1 control. Whereas the pGEX-CdgB$_{EAL-AAL}$ strain increased the accumulation of red fluorescent bacterial cells and colonies (**Fig 8**). These results strongly suggest that CdgB has both DGC and PDE activities in *E. coli*.

## CdgB is potentialy polarly localized

Studies have shown that some proteins involved in c-di-GMP metabolism need an specific localization to regulate cell behavior [60–62]. Bioinformatic analyses predicted that CdgB contains an MHYT domain with seven transmembrane helices (**Fig 1**). These observations prompted us to evaluate if CdgB is anchored to the cell membrane through its MHYT domain. To do so, we analyzed the localization of a CdgB-eGFP fusion protein through fluorescence microscopy. To better delineate the cellular membrane, we stained the cells with a membrane lipid-specific dye (FM4-64FX). We observed CdgB anchored to the cell membrane and eventually arrive to cell poles (**Fig 9**) (S1 Video). The membrane anchor of CdgB depended on the presence of the MHYT domain (**Fig 9**).

We also confirmed through immunoblot analysis that the strain of *A. baldaniorum* expressing CdgB$_{ΔMHYT}$ produced a truncated CdgB-eGFP protein of 85 kDa in the cytoplasmic fraction, while the *A. baldaniorum* strain expressing CdgB-eGFP produced a protein of 112 kDa, mainly detected in protein extracts obtained by detergent solubilization (**S4 Fig**). Our results unveiled that CdgB is possibly polarly localized. We found, in both strains tested (*A. baldaniorum*+*cdgB*::*egfp* and *A. baldaniorum* Δ*cdgB*+*cdgB*::*egfp*), that CdgB-eGFP displayed unipolar, bipolar and multisite distributions (**S5 Fig**). This compartmentalization could play a role in controlling the activity of this bifunctional enzyme perhaps through a signal sensed by the MHYT domain in the periplasm.

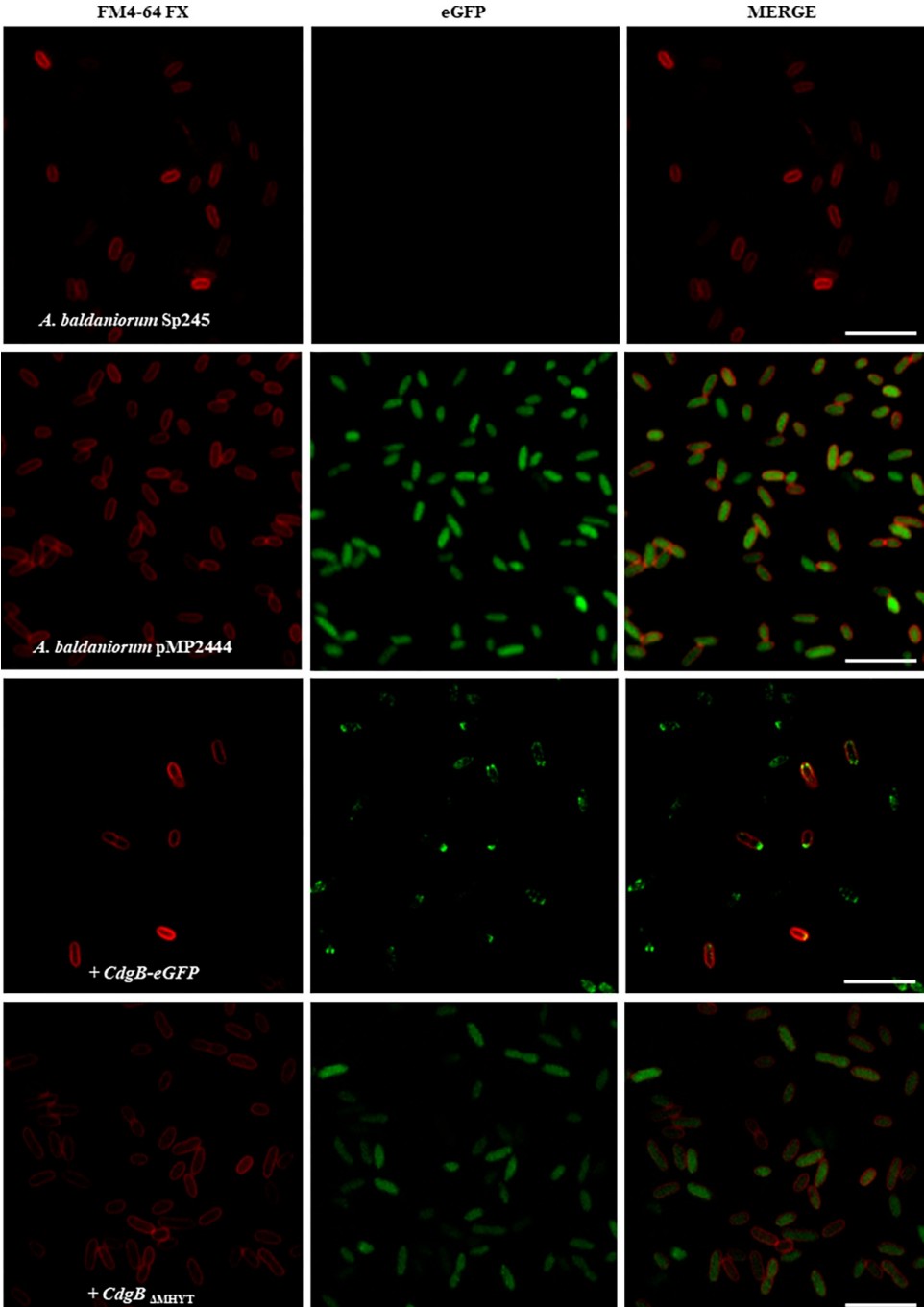

**Fig 9. CdgB is anchored to the cytoplasmic membrane and polarly localized.** The first two columns (left to right) show representative images acquired to detect fluorescence of the FM4-64FX fluorophore (red) that bind to lipidic membranes, and fluorescence from the eGFP protein (green). The third column shows images where the two fluorescent signals were merged. The white arrowheads indicate polar localization of CdgB protein. The white bar corresponds to 10 µm.

## Discussion

*A. baldaniorum* Sp245 is an environmental bacterium capable of establishing mutualistic relationships with a variety of plants. Both as a soil dwelling bacterium and a symbiont, *A. baldaniorum* Sp245 is exposed to a plethora of signals that need to be integrated by sensory modules to adapt its behavior and increase fitness. Signaling modules that incorporate the second messenger c-di-GMP are crucial for this bacterium to engage in sessile or motile lifestyles [2, 17–19, 63]. Here we reported our findings regarding the characterization of a GGDEF-EAL protein with PAS and MHYT sensory domains which we named CdgB. Proteins with GGDEF-EAL tandem domains can be grouped based on the conservation of the GGDEF and EAL motifs. Some of these proteins have only one of the two motifs conserved [47], both domains degenerated [64, 65], or both motifs conserved [10, 47, 66]. Typically, even when both motifs are conserved, these proteins have a predominant catalytic activity. Evidence has been accumulating of the role of bifunctional GGDEF-EAL proteins with dual enzymatic activity (DGC and PDE). The predominant activity of these proteins is controlled by self-made or external signals [67–69]. Our results strongly suggest that CdgB is a bifunctional enzyme. In the genetic background of *A. baldaniorum* Sp245, under the conditions tested, overproduction of CdgB results in phenotypic changes typically associated with the activity of PDEs. Interestingly, in a heterologous host the predominant activity observed was of DGC. These differences could stem from signals that are species-specific or the result of the growth conditions used for these experiments. We speculate that the different behavior of CdgB in *E. coli* could be due to differences in nitrogen metabolism. Recently Park et al. [70] demonstrated that a nitrite transporter stimulated biofilm formation by controlled NO production via appropriate nitrite reductase suppression and subsequent diguanylate cyclase activation, and c-di-GMP production in *E. coli* and *P. aeruginosa*. Additionally, *E. coli* possesses a transcription factor named NorR with a non-heme iron center to sense NO [71], indication of different regulation mechanisms. More work needs to be done to elucidate what could be favoring the DGC activity of CdgB in *E. coli*.

The sensory domain MHYT could potentially play an important role in controlling the enzymatic activity of CdgB. The first characterized hybrid protein DGC-PDE with an MHYT domain was YkoW from *Bacillus subtilis* although the involvement of this domain in signal perception and enzymatic function could not be ascertained. Nevertheless, it was proposed that the MHYT domain could potentially sense $O_2$, CO, or NO through the coordination of one or two copper atoms [46]. Other proteins characterized to date with an MHYT domain are NbdA and MucR from *P. aeruginosa* both implicated in NO-mediated biofilm dispersion. The deletion of *nbdA* and *mucR* impaired NO-induced biofilm dispersion, only NbdA appeared to be specific to the NO-induced dispersion response. Although MucR was shown to play a role in biofilm dispersion in response to NO and glutamate, exposure to NO in the presence of MucR but the absence of NbdA did not result in increased PDE activity. MucR is one of the few proteins harboring both EAL and GGDEF domains that possess both DGC and PDE activity [47, 66]. Furthermore, MucR is proposed to interact with the alginate biosynthesis protein Alg44, which contains a c-di-GMP binding PilZ domain essential for alginate biosynthesis [5, 66]. Recently, other MucR homologous protein was described in *Azotobacter vinelandii* implicated in the alginate biosynthesis too [72].

Oxygen and nitric oxide are key signals during the establishment of *A. baldaniorum* Sp245 as a symbiont [2, 40]. Interestingly, there are at least two chemotaxis receptors, Tlp1 and Aer, involved in $O_2$ sensing that incorporate c-di-GMP detection through a conserved PilZ domain [2, 4]. Aer also integrates other chemical cues produced by the roots [4]. These chemosensory receptors could be potential signaling partners of CdgB which could perhaps sense similar chemotaxis signals. The PAS domain of CdgB could add another layer to its sensory repertoire.

This domain is present in a variety of bacterial signaling proteins; and is able to bind several molecules, such as oxygen and Flavin adenine dinucleotide (FAD) [9, 73]. NO has been shown to promote root growth promotion [74] and biofilm formation in *A. baldaniorum* Sp245, the absence of a periplasmic nitrate reductase (Nap) significantly affects biofilm formation through a mechanism yet to be explored [40]. Since the MHYT domain has been proposed to sense NO it is tempting to speculate that CdgB could participate in a singling module that controls biofilm formation in response to nitric oxide. We speculate that NO may be sensed by the MHYT domain of CdgB. Reception of this signal may influence CdgB activity by modulating DGC activity and possibly PDE domain activity. Nevertheless, the molecular details of this mechanism remain largely unknown, and need more analyses.

The polar localization of CdgB is intriguing. It remains to be shown if this recruitment occurs in the flagellated cell pole and who are the interacting partners of CdgB. Cellular compartmentalization of signaling modules opens the possibility of localized sensing and short-range signal transduction to a closely localized effector. Future work will be aimed to identify potential members of the CdgB signaling module.

## Supporting information

**S1 Fig. Genetic context and MHYT alignment of CdgB.** (A) Genetic context of the *cdgB* gene (AZOBR_410089) which codifies the CdgB protein is composed of a putative *lysR* gene upstream (AZOBR_410088) and a gene of unknown function downstream (AZOBR_410090). (B) Schematic representation of the domain structures of CdgB. The numbers represent the start and end amino acid of the predicted domains based on SMART database. (C) The Sequence alignment of the MHYT domain of CdgB with other c-di-GMP metabolizing proteins harboring an MHYT domain: the YkoW protein of *B. subtilis*, NbdA and MucR of *P. aeruginosa*. The three MHYT motifs are labeled by black squares and the transmembrane regions determined by the SMART and TOPCONS programs are highlighted in red.
(TIF)

**S2 Fig. Phylogenetic analysis of *cdgB* gene.** Phylogenetic tree showing the *cdgB* gene of *A. baldaniorum* Sp245 conserved in the alphaproteobacteria class. Nucleotide sequences of *cdgB* gene and homologous related genes were aligned by Clustal Omega. Phylogenetic tree files were generated by MEGA version VII. The phylogenetic tree was inferred using the Maximum Likelihood method and Tamura-Nei substitution model, with 1000 bootstrap replications.
(TIF)

**S3 Fig. Growth curve of *A. baldaniorum* Sp245, *A. baldaniorum* pMP2444, *A. baldaniorum* +pMP-*cdgB*, *A. baldaniorum* Δ*cdgB*, and *A. baldaniorum* Δ*cdgB* +pMP-*cdgB*.** The strains were grown in NFB* media. Growth was measured at $OD_{600}$ nm every 2–3 hours. Cultures were kept for 24 h at 30°C under agitation (150 rpm). Data showed is representative from three independent cultures of each strain.
(TIF)

**S4 Fig. Western blot detection of CdgB-eGFP and ΔMHYT-CdgB-eGFP fusions.** Detection by Western blotting of CdgB::eGFP fusion proteins from soluble and detergent treatment protein extracts of *A. baldaniorum cdgB*::*egfp* strain, and the truncated protein variant from the ΔMHYT-*cdgB*::*egfp* derivative strain, harboring the plasmid pMP+*cdgB*$_{ΔMHYT}$. The GFP D5.1 Rabbit mAb, HRP Conjugate against GFP was used. Lane 1 *A.baldaniorum cdgB*::*egfp*; lane 2 ΔMHYT-*cdgB*, variant protein; lane 3. *A. baldaniorum* pMP2444. The relative molecular weight (MMr) is indicated as kDa. The black arrows indicate MMr in kDa. The corresponding

figures represent the proteins indicate in each well.
(TIF)

**S5 Fig. Subcellular localization of CdgB in *A. baldaniorum* strains. A)** *A. baldaniorum+ cdgB*::*egfp*, **B)** *A. baldaniorum* Δ*cdgB*+*cdgB*::*egfp* were grown in NFB* medium. Photographs of the GFP fusion protein in WT and Δ*cdgB* strains were detected using a fluorescence microscope (TE 2000U; Nikon). Different subcellular locations of protein CdgB-eGFP including polar, bipolar, multisite were visualized. In red is showed membrane lipids with FM4-64FX, in green the CdgB-eGFP fusion, and blue bacterial nucleoid staining with DAPI. The images are representative of three biology repeats. Scale bar correspond to 10 μm.
(TIF)

**S1 Video. Visualization of *A. baldaniorum cdgB*::e*gfp*. Time-lapse.** The samples were scanned at an x/y scanning resolution of 1,024 x 1,024 pixels. Step size in z directions was 0.05 μm. The Plan Apo VC 60X WI objective was used. The eGFP was excited at 488nm, and the FM4-64FX was excited at 561 nm with objective lens WI (water immersion).
(AVI)

## Acknowledgments

We wish to thank Martha Minjárez Saénz for initial technical assistance (Structural analysis of CdgB).

## Author Contributions

**Conceptualization:** Víctor I. Viruega-Góngora, Beatriz Eugenia Baca, Alberto Ramírez-Mata.

**Data curation:** Víctor I. Viruega-Góngora, Iris S. Acatitla-Jácome.

**Formal analysis:** Víctor I. Viruega-Góngora, María L. Xiqui-Vázquez, Alberto Ramírez-Mata.

**Funding acquisition:** Beatriz Eugenia Baca, Alberto Ramírez-Mata.

**Investigation:** Víctor I. Viruega-Góngora, Iris S. Acatitla-Jácome, David Zamorano-Sánchez, Sandra R. Reyes-Carmona, María L. Xiqui-Vázquez, Beatriz Eugenia Baca, Alberto Ramírez-Mata.

**Methodology:** Víctor I. Viruega-Góngora, Iris S. Acatitla-Jácome, Sandra R. Reyes-Carmona, María L. Xiqui-Vázquez.

**Project administration:** María L. Xiqui-Vázquez, Beatriz Eugenia Baca, Alberto Ramírez-Mata.

**Supervision:** David Zamorano-Sánchez, Sandra R. Reyes-Carmona, Beatriz Eugenia Baca, Alberto Ramírez-Mata.

**Validation:** Sandra R. Reyes-Carmona, Beatriz Eugenia Baca.

**Visualization:** Alberto Ramírez-Mata.

**Writing – original draft:** Víctor I. Viruega-Góngora.

**Writing – review & editing:** David Zamorano-Sánchez, Beatriz Eugenia Baca, Alberto Ramírez-Mata.

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
