## [Decision Letter · Decision Letter 0]

11 Sep 2022

PONE-D-22-21548The GGDEF-EAL protein CdgB from Azospirillum baldaniorum Sp245, is a dual function enzyme with potential polar localizationPLOS ONE

Dear Dr. Ramírez-Mata,

Thank you for submitting your manuscript to PLOS ONE. After careful consideration, we feel that it has merit but does not fully meet PLOS ONE’s publication criteria as it currently stands. Therefore, we invite you to submit a revised version of the manuscript that addresses the points raised during the review process, specifically the below items. 

Clarification of cdgB was cloned along with its native promoter or under the plac

A quantification of cellular c-di-GMP level in A. baldanorium strains

Explain clearly on what is the difference between CdgB and previously reported GGDEF-EAL proteins in the discussion 

Predict the function of CdgB in the mutualistic relationships between Azospirillum baldaniorum and plants Please submit your revised manuscript by Oct 26 2022 11:59PM. If you will need more time than this to complete your revisions, please reply to this message or contact the journal office at plosone@plos.org. Please include the following items when submitting your revised manuscript:A rebuttal letter that responds to each point raised by the academic editor and reviewer(s). You should upload this letter as a separate file labeled 'Response to Reviewers'.A marked-up copy of your manuscript that highlights changes made to the original version. You should upload this as a separate file labeled 'Revised Manuscript with Track Changes'.An unmarked version of your revised paper without tracked changes. You should upload this as a separate file labeled 'Manuscript'.

We look forward to receiving your revised manuscript.

Kind regards,

Ching-Hong Yang

Academic Editor

PLOS ONE

Journal Requirements:

"Include this sentence at the end of your statement: The funders had no role in study design, data collection and analysis, decision to publish, or preparation of the manuscript."

"VIVG received a scholarship from the Consejo Nacional de Ciencia y Tecnología de Mexico (CONACYT). This work was financially supported by CONACYT, number grant: INFR-2014-01-225923), Vicerrectoria de Investigación y Estudios de Posgrado (VIEP)."

"Include this sentence at the end of your statement: The funders had no role in study design, data collection and analysis, decision to publish, or preparation of the manuscript."

6. Please upload a new copy of Figures 2, 3, and 7 as the detail is not clear. Please follow the link for more information: https://blogs.plos.org/plos/2019/06/looking-good-tips-for-creating-your-plos-figures-graphics/

https://blogs.plos.org/plos/2019/06/looking-good-tips-for-creating-your-plos-figures-graphics/

Reviewers' comments:

Reviewer's Responses to Questions

**Comments to the Author**

1. Is the manuscript technically sound, and do the data support the conclusions?

Reviewer #1: Partly

Reviewer #2: Yes

2. Has the statistical analysis been performed appropriately and rigorously? 

Reviewer #1: No

Reviewer #2: Yes

3. Have the authors made all data underlying the findings in their manuscript fully available?

Reviewer #1: Yes

Reviewer #2: Yes

4. Is the manuscript presented in an intelligible fashion and written in standard English?

Reviewer #1: Yes

Reviewer #2: Yes

5. Review Comments to the Author

Reviewer #1: The manuscript requires a few more experimentation and explanations to some of the aspects discussed. Statistical analysis between samples is unclear and needs revision. The graphs with bars and T-test analysis and their legends are not enough explanatory, and requires minor changes. Please see the specific comments attached. Revisions would significantly benefit the manuscript for the readers.

Reviewer #2: The manuscript by Alberto Ramírez-Mata et al entitled “The GGDEF-EAL protein CdgB from Azospirillum baldaniorum Sp245, is a dual function enzyme with potential polar localization” is well written and presented. Experiments appear to have been well performed and results clear. The authors studied the dual activity of CdgB, a predicted MHYT-PAS-GGEDF-EAL multidomain protein from Azospirillum baldaniorum.

The manuscript will be improved if the authors state state more clearly what is the difference between CdgB and previously reported GGDEF-EAL proteins in discussion. This was metioned in line 494-500 but in my opinion, this didn’t go far enough.

In addition, could the author predict the function of CdgB in the mutualistic relationships between Azospirillum baldaniorum and plants?

6. PLOS authors have the option to publish the peer review history of their article (what does this mean?). If published, this will include your full peer review and any attached files.

Reviewer #1: No

Reviewer #2: **Yes: **Fan Susu

---

## [Author Response · Author response to Decision Letter 0]

24 Oct 2022

Minor comments

-- Line 26: change “stablishing” to establishing. Corrected

-- Line 27: in silico italicize. Corrected

-- Line 30: Sp245. It inhibits… Corrected

-- Line 34: MHYT domain negatively affects extracellular… Corrected

-- Line 37: under certain conditions… Corrected

-- Line 45: plays a central role in microbial cellular behaviors in response to the environment… Corrected

-- Line 53: delete “,” after intra-. Corrected (Line 54)

-- Line 59: Bioinformatic. Corrected (Line 60)

-- Line 81: CdgB DGC and PDE activity under same conditions or different? Please mention. Corrected (Line 82)

-- Line 85: What is PGPR? First time used abbreviation should be elaborated. Corrected (Line 87)

-- Line 113: Gentamicin. Corrected (Line 114)

-- Line 245-46: How did you measure the color intensity? Please mention. Corrected, we added the following text: The relative fluorescence intensity (RFI) was calculated as the ratio between the TurboRFP and AmCyan fluorescence intensities and is directly proportional to c-di-GMP levels, as analyzed using ImageJ software. The RFI values represent the standard deviations of three biological replicates, and significant differences are indicated at * P < 0.05 according to Student´s t-test by SigmaPlot as previously described (Lines 257-261)

-- Line 311: proposed to be… Corrected (Line 320)

-- Line 449: riboswitches? Or riboswitch? Please specify one or multiple. Corrected (Line 242)

-- Line 508: Bacillus subtilis, although… Corrected (Line 594)

Major comments

Request 1

-- Line 144: For overexpression of cdgB was cloned along with its native promoter? Or the ORF of the gene was cloned in frame under the plac? If the native promoter was included, then there is a possibility of additional regulation at the cdgB promoter. The phenotypes observed in A. baldaniorum might include additional effects of cdgB transcriptional regulation. The experiments with overexpression of cdgB will benefit from constitutive expression under lac promoter. 

Response 1

cdgB gene overexpression is under the control of the lac promoter only. We corrected the mistake. The full-length ORF of cdgB gene (Line 145).

Request 2

Additional suggestion: to support the expression of all the native and mutated variants of CdgB, a western blot of tagged protein would be strong evidence for full expression of the protein variants. This would confirm that the phenotypes observed was not because of any artifacts.

Response 2

We agree that without the requested evidence, it is not possible to discard potential stability effects of introducing point mutations into cdgB. Since we do not have specific antibodies to detect these proteins it would not be possible to provide these data in time for the revision deadline. However, multiple reports have used the same type of point mutations in DGC-PDE proteins with no effect on stability. We did analyze the stability of the fusions CdgB-eGFP full-length and CdgBΔMHYT-eGFP by western blot using an eGFP antibody (S4 figure) (Lines 205, 416-417) and did not observed differences between them. In the revised version of the manuscript we show that the fusion protein expressed from pMP-cdgB::egfp is functional in biofilm and Congo Red assays (figures 4 and 5). 

Request 3

-- Fig 4, 5 and 7: Biofilm, EPS production and swimming assay- the T test and the annotations on the graph for statistical significance is confusing and doesn’t compare all the samples tested. A One-way ANOVA followed by a multiple comparison test (Like Tukey’s test) would actually compare all the sample sets and give the readers a clear idea of the statistically significant differences among each of the samples. Also, was T-test performed between each of the samples or just between each control and their corresponding tests samples? Need a mention. The asterisks and the labels above the bars in the graphs are not clear which samples are being compared. This can be explained briefly in figure legends.

Response 3

In response to the reviewer comments, we remade the plots from figures 4, 5 and 7. We deleted the marks on plots and compared each control with their corresponding tests samples. 

Request 4

-- The c-di-GMP differences in various strains and overexpression of cdgB was measured mostly by qualitative methods. A quantitative measurement of cellular c-di-GMP would be helpful to understand the dominance of either DGC or PDE activity of CdgB under different conditions. 

Response 4

In response to the reviewer comments, in the revised version of the manuscript we provide quantitative values for the results obtained using the c-di-GMP reporter (See figure 8). To quantify the fluorescence of cells from fluorescent images, we used a previously reported method described by (Cruz-Pérez et al 2021 and Shihan M et al 2021). The excitation and emission settings for the detection of Amcyan and TurboRFP were 457/520 nm and 553/574 nm, respectively. The relative fluorescence intensity (RFI) was calculated by dividing the arbitrary fluorescent intensity units of TurboRFP by those of Amcyan as previously described (Cruz-Pérez et al 2021; Shihan MH et al 2021.) Experiments were repeated at least three times.

References

Cruz-Pérez JF, Lara-Oueilhe R, Marcos-Jiménez C, Cuatlayotl-Olarte R, Xiqui-Vázquez ML, Reyes-Carmona SR, Baca BE, Ramírez-Mata A. Expression and function of the cdgD gene, encoding a CHASE-PAS-DGC-EAL domain protein, in Azospirillum brasilense. Sci Rep. 2021 Jan 12;11(1):520. doi: 10.1038/s41598-020-80125-3. PMID: 33436847; PMCID: PMC7804937.

Shihan MH, Novo SG, Le Marchand SJ, Wang Y, Duncan MK. A simple method for quantitating confocal fluorescent images. Biochem Biophys Rep. 2021 Feb 1;25:100916. doi: 10.1016/j.bbrep.2021.100916. PMID: 33553685; PMCID: PMC7856428.

Request 5

-- A quantification of cellular c-di-GMP level in A. baldanorium strains is required. The differences of phenotypes (like biofilm, swimming, etc.) observed in the strains because of deletion or overexpression of cdgB is caused by variation of c-di-GMP level, needs to be established. 

Response 5

We agree it is important to optimize a methodology to quantify c-di-GMP levels in A. baldinorium. We could not use the same fluorescent c-di-GMP biosensor that allowed us to detect c-di-GMP levels in E. coli because the plasmid used for overexpression of cdgB in A. baldinorium is of the same incompatibility group as the plasmid that harbors the c-di-GMP biosensor. We are beginning the quantitative standardization of c-di-GMP level by HPLC coupled to mass spectrometry. In a future work we are planning to measure c-di-GMP levels in A. baldanorium cells grown under different conditions, such as presence and absence of NO, O2, CO2 and other type of signals. 

Request 6

-- The DGC activity of CdgB increased by overexpression in E. coli but the reverse in A. baldanorium. Is it host specific? Requires an elaboration in the discussion section. A quantitative measurement of cellular c-di-GMP can be done with overexpression of cdgB in both A. baldanorium and E. coli under the same or similar condition and a comparison can be shown to determine the concentration-dependent catalytic activity of CdgB. Then conclusion can be made whether the DGC activity of CdgB in E. coli is host-specific or conditional. 

Response 6

We consider the observation that CdgB acts mainly as a DGC in E. coli as evidence of its capability of dual activity. The factors that affect the activity switch will be explored in future work. In A. baldanorium it appears that the abundance of CdgB could affect the DGC-PDE switch. The absence of CdgB results in a decrease in biofilm formation, a phenotype that is positively influenced by c-di-GMP. This would suggest that at baseline levels CdgB may act as a DGC. When cdgB is overexpressed in a genetic background that has a native copy of cdgB, biofilm formation is also inhibited. This would suggest that in these conditions CdgB has switched from DGC to PDE activity. Interestingly, when a variant with a putative dead PDE domain was overproduced biofilm formation was significantly induced. This strongly suggests that CdgB can switch between the two activities in A. baldanorium. We speculate that the different behavior of CdgB in E. coli could be due to differences in nitrogen metabolism. Recently Park et al 2020 demonstrated that a nitrite transporter stimulated biofilm formation by controlled NO production via appropriate nitrite reductase suppression and subsequent diguanylate cyclase activation, and c-di-GMP production in E. coli and P. aeruginosa. Additionally, E. coli possesses a transcription factor named NorR with a non-heme iron center to sense NO (D'Autréaux B, et al 2005), indication of different regulation mechanisms. More work needs to be done to elucidate what could be favoring the DGC activity of CdgB in E. coli (Lines 583-591)

Reference

Park JS, Choi HY, Kim WG. The Nitrite Transporter Facilitates Biofilm Formation via Suppression of Nitrite Reductase and Is a New Antibiofilm Target in Pseudomonas aeruginosa. mBio. 2020 Jul 7;11(4):e00878-20. doi: 10.1128/mBio.00878-20. PMID: 32636243; PMCID: PMC7343986.

D'Autréaux B, Tucker NP, Dixon R, Spiro S. A non-haem iron centre in the transcription factor NorR senses nitric oxide. Nature. 2005 Sep 29;437(7059):769-72. doi: 10.1038/nature03953. PMID: 16193057.

Request 7

-- Deletion of cdgB resulted in a different phenotype compared to overexpression in A. baldanorium. This suggests that high cellular concentration of CdgB increases its PDE activity in A. baldanorium. This means that increased transcription and subsequently translation of cdgB is responsible for reduced c-di-GMP in the bacterium in response to the environment. An experiment showing transcription of cdgB and also its translation under different conditions such as presence and absence of NO, O2 and CO2 followed by measurement of intracellular c-di-GMP levels in A. baldanorium. It can be beneficial and complement the data shown in the manuscript. This experiment can be used to determine if the PDE and DGC activity of CdgB is dependent on its intracellular concentration or environmental signals, since the signals perceived by this protein is to be determined in future. 

Response 7

The reviewer’s suggestion is very interesting. Our goal for this manuscript is to be the anchor point for future analysis. The mater of the regulation of cdgB is one of our current interests. It is unclear if cdgB has its own promoter upstream or if it is co-transcribed with its neighbor genes, a putative lysR gene upstream and a gene of unknown function downstream (see S1 Figure) (Lines 892-894). Together with an in-depth characterization of the regulatory region of cdgB, as the reviewer suggested, we are looking to analyze how its expression is affected by signals that could be detected by the MHYT domains, such as NO, O2 and CO2. 

Request 8

Explain clearly on what is the difference between CdgB and previously reported GGDEF-EAL proteins in the discussion.

The first characterized hybrid protein DGC-PDE with an MHYT domain was YkoW from Bacillus subtilis although the involvement of this domain in signal perception and enzymatic function could not be ascertained. Nevertheless, it was proposed that the MHYT domain could potentially sense O2, CO, or NO through the coordination of one or two copper atoms [46]. Other proteins characterized to date with an MHYT domain are NbdA and MucR from P. aeruginosa both implicated in NO-mediated biofilm dispersion. The deletion of nbdA and mucR impaired NO-induced biofilm dispersion, only NbdA appeared to be specific to the NO-induced dispersion response. Although MucR was shown to play a role in biofilm dispersion in response to NO and glutamate, exposure to NO in the presence of MucR but the absence of NbdA did not result in increased PDE activity. MucR is one of the few proteins harboring both EAL and GGDEF domains that possess both DGC and PDE activity [47,67]. Furthermore, MucR is proposed to interact with the alginate biosynthesis protein Alg44, which contains a c-di-GMP binding PilZ domain essential for alginate biosynthesis [5,67]. Recently, other MucR homologous protein was described in Azotobacter vinelandii implicated in the alginate biosynthesis too [73]. (Lines 597-608)

Request 9

Predict the function of CdgB in the mutualistic relationships between Azospirillum baldaniorum and plants.

NO has been shown to promote root growth promotion[75] and biofilm formation in A. baldaniorum Sp245, the absence of a periplasmic nitrate reductase (Nap) significantly affects biofilm formation through a mechanism yet to be explored [40]. Since the MHYT domain has been proposed to sense NO it is tempting to speculate that CdgB could participate in a singling module that controls biofilm formation in response to nitric oxide. We speculate that NO may be sensed by the MHYT domain of CdgB. Reception of this signal may inﬂuence CdgB activity by modulating DGC activity and possibly PDE domain activity. Nevertheless, the molecular details of this mechanism remain largely unknown, and need more analyses. (Lines 617-625)

---

## [Decision Letter · Decision Letter 1]

9 Nov 2022

The GGDEF-EAL protein CdgB from Azospirillum baldaniorum Sp245, is a dual function enzyme with potential polar localization

PONE-D-22-21548R1

Dear Dr. Ramírez-Mata,

We’re pleased to inform you that your manuscript has been judged scientifically suitable for publication and will be formally accepted for publication once it meets all outstanding technical requirements.

Kind regards,

Ching-Hong Yang

Academic Editor

PLOS ONE

Additional Editor Comments (optional):

Reviewers' comments:

Reviewer's Responses to Questions

**Comments to the Author**

1. If the authors have adequately addressed your comments raised in a previous round of review and you feel that this manuscript is now acceptable for publication, you may indicate that here to bypass the “Comments to the Author” section, enter your conflict of interest statement in the “Confidential to Editor” section, and submit your "Accept" recommendation.

Reviewer #1: All comments have been addressed

Reviewer #2: All comments have been addressed

2. Is the manuscript technically sound, and do the data support the conclusions?

Reviewer #1: Yes

Reviewer #2: Yes

3. Has the statistical analysis been performed appropriately and rigorously? 

Reviewer #1: Yes

Reviewer #2: Yes

4. Have the authors made all data underlying the findings in their manuscript fully available?

Reviewer #1: Yes

Reviewer #2: Yes

5. Is the manuscript presented in an intelligible fashion and written in standard English?

Reviewer #1: Yes

Reviewer #2: Yes

6. Review Comments to the Author

Reviewer #1: The authors have sufficiently addressed all the concerns in their response to reviewers. I believe the manuscript is in a better shape than before.

Reviewer #2: (No Response)

7. PLOS authors have the option to publish the peer review history of their article (what does this mean?). If published, this will include your full peer review and any attached files.

Reviewer #1: No

Reviewer #2: **Yes: **Fan Susu

---

## [Editor Report · Acceptance letter]

14 Nov 2022

PONE-D-22-21548R1 

The GGDEF-EAL protein CdgB from *Azospirillum baldaniorum* Sp245, is a dual function enzyme with potential polar localization. 

Dear Dr. Ramírez-Mata:

I'm pleased to inform you that your manuscript has been deemed suitable for publication in PLOS ONE. Congratulations! Your manuscript is now with our production department. 

Kind regards, 

on behalf of

Dr. Ching-Hong Yang 

Academic Editor

PLOS ONE